# Test-time Data Augmentation for Estimation of Heteroscedastic Aleatoric Uncertainty in Deep Neural Networks

**Murat Seçkin Ayhan and Philipp Berens**
Institute for Ophthalmic Research
University of Tübingen
`{murat-seckin.ayhan, philipp.berens}@uni-tuebingen.de`

## Abstract

Deep neural networks (DNNs) have revolutionized medical image analysis and disease diagnosis. Despite their impressive increase in performance, it is difficult to generate well-calibrated probabilistic outputs for such networks such that state-of-the-art networks fail to provide reliable *uncertainty* estimates regarding their decisions. We propose a simple but effective method using traditional data augmentation methods such as geometric and color transformations at test time. This allows to examine how much the network output varies in the vicinity of examples in the input spaces. Despite its simplicity, our method yields useful estimates for the input-dependent predictive uncertainties of deep neural networks. We showcase the impact of our method via the well-known collection of fundus images obtained from a previous Kaggle competition.

## 1 Introduction

Deep neural networks (DNNs) have emerged as powerful image analysis and prediction tools also in medical image analysis and disease diagnosis. For instance, DNNs surpassed or achieved human-level performance on skin cancer classification from dermoscopic images [1] and diabetic retinopathy (DR) detection from fundus images [2] . Despite their impressive improvement in various performance metrics, such as accuracy, sensitivity, specificity, F1 score, or ROC-AUC, which mainly describe a model's discriminative power, DNNs do not generate well-calibrated reliable *uncertainty* estimates regarding their decisions [3, 4, 5, 6]. Especially in medical settings, uncertainty estimates are crucial, however [7].

The predictive uncertainty of neural networks can be decomposed into two parts: *epistemic uncertainty* and *aleatoric uncertainty* [5]. Epistemic uncertainty can be formalized by means of a probability distribution over the model parameters and accounts for our ignorance about them. It is also known as model uncertainty and can be explained away given enough data [5]. The remaining uncertainty is called aleatoric uncertainty and described by the noise inherent in observations, that is the model's input-dependent uncertainty. Despite the epistemic uncertainty may vanish in the case of zero parameter ignorance, aleatoric uncertainty remains, for instance, due to sensor noise and motion noise. Aleatoric uncertainty cannot be explained away with more data; however, it can be formalized by a distribution over model outputs [5]. Aleatoric uncertainty can be further divided into two subtypes: *homoscedastic* and *heteroscedastic*. In the case of homoscedasticity, uncertainty is assumed to be constant for different inputs. Heteroscedastic uncertainty is useful when modeling assumptions include variable noise across the parts of input space. Considering the epistemic and aleatoric uncertainties as the properties of model and data, respectively, [5] proposed a Bayesian deep learning framework that jointly models both types of uncertainties. While simultaneous modeling of both enables the best efficacy, the overall performance is dominated by the explanation of aleatoric

1st Conference on Medical Imaging with Deep Learning (MIDL 2018), Amsterdam, The Netherlands.

uncertainty. The contribution from the epistemic component is marginal. Moreover, the explanation of aleatoric uncertainty compensates for the epistemic uncertainty when they are modeled separately. Therefore, in the regime of big data, it is more effective to tackle aleatoric uncertainty, since the epistemic one will be explained away. Moreover, "we can form aleatoric models without expensive Monte Carlo samples." - [5, p.9], which generously opens up new avenues to explore.

Here we propose to model aleatoric uncertainty based on data augmentation techniques. The goal of data augmentation is to obtain additional examples for a dataset so that a machine learning algorithm can learn from a larger and preferably better set of examples. Although the data augmentation strategy for training is well-established and has been used to improve the discriminative performance of models during inference, to the best of our knowledge, estimation of predictive uncertainty via test-time data augmentation is an unexplored territory. We focus on traditional data augmentation methods, such as simple geometric and color transformations. We hypothesize that such simple but random transformations of examples will enable us to capture heteroscedastic aleatoric uncertainty of deep neural networks. In this setting, we emphasize the aleatoric uncertainty; however, data augmentation during training may also help reduce the epistemic uncertainty since it contributes to learning experience of networks.

In the next section, we review the recent methods for the estimation of predictive uncertainties of deep neural networks. Then, we present ours and demonstrate its impact on a well-known diabetic retinopathy (DR) detection dataset. Lastly, we discuss the results and provide an outlook for further evaluation of the test-time data augmentation strategy for predictive uncertainty estimation.

## 2   Related Work

Despite the fact that Bayesian neural networks are easy to formulate, they are computationally intractable for many real world problems [3, 5, 8]. In this regard, *dropout* [9], which is essentially a regularization technique based on random selections of active neurons, has been used to perform approximate Bayesian inference [3]. By turning dropout on at test time and having $T$ forward passes, given a test case, one can obtain a predictive distribution as a proxy for the true predictive posterior. This method is called *Monte Carlo Dropout (MCDO)*. MCDO is simple, efficient and captures model uncertainty. Leibig et al. used a VGG-like CNN equipped with MCDO for DR detection from fundus images [7] . To mimic a clinical work flow, they leveraged the predictive uncertainty information and referred difficult cases for further inspection, which resulted in substantial improvements in DR detection performance in the remaining data.

Recently, *Batch Normalization (BN)* has been cast as an approximate Bayesian inference method: *Monte Carlo Batch Normalization (MCBN)* [10]. Essentially, BN [11, 12] uses the lower order moments ($\mu_{\mathcal{B}}, \sigma_{\mathcal{B}}$ in Eq. 1) of *pre*/activations ($\mathbf{x}$) generated by a neuron, given a minibatch of size $m$ sampled during stochastic training, and normalizes them w.r.t. the minibatch statistics. Then, it restores the representational power of the network via an affine transform that scales ($\gamma$) and shifts ($\beta$) the normalized values ($\widehat{x}_i$). Note that $\gamma$ and $\beta$ are learned via backpropagation [11].

$$\mu_{\mathcal{B}} = \frac{1}{m} \sum_{i=1}^{m} x_i \qquad \sigma_{\mathcal{B}} = \sqrt{\frac{1}{m} \sum_{i=1}^{m} (x_i - \mu_{\mathcal{B}})^2}$$
$$\widehat{x}_i = \frac{x_i - \mu_{\mathcal{B}}}{\sigma_{\mathcal{B}}} \qquad y_i = \gamma \widehat{x}_i + \beta \equiv \mathrm{BN}_{\gamma,\beta}(x_i)$$
(1)

BN reduces the network's internal covariate shift and stabilizes the training of neural networks. Also, minibatch statistics collected during training give rise to moving averages $\mu$ and $\sigma$ that are used during inference. Ultimately, BN improves the generalization performance of neural networks. As a by product, it offers a compelling regularization performance [11, 12]. Thus, BN has replaced dropout and state-of-the-art networks, e.g., ResNets [13, 14], use only BN. However, BN has its drawbacks [12]: i) the estimates of $\mu_{\mathcal{B}}$ and $\sigma_{\mathcal{B}}$ become inaccurate and the effectiveness of BN diminishes, when $m$ is small, ii) non-i.i.d. examples in minibatches interact at every layer, which may cause the network to overfit to the specific distribution of minibatches, iii) the activities generated by the network during training and inference may differ. *Batch Renormalization (BReN)* [12] reinforces BN by relating the minibatch statistics to the moving averages via an affine transformation (Eq. 2) and corrects them

while training. Thus, it uses more informed statistics even if $m$ is small, addresses the discrepancy between the activations generated during training and inference, and improves on BN.

$$\frac{x_i - \mu}{\sigma} = \frac{x_i - \mu_{\mathcal{B}}}{\sigma_{\mathcal{B}}} r + d, \text{ where } r = \frac{\sigma_{\mathcal{B}}}{\sigma}, d = \frac{\mu_{\mathcal{B}} - \mu}{\sigma} \tag{2}$$

MCBN exploits the stochasticity of minibatch statistics in order to obtain predictive distributions. $T$ forward passes with random minibatches sampled from training data during inference leads to a predictive distribution for a test case. In this respect, MCBN also captures model uncertainty. However, MCBN explicitly shies aways from the use of moving averages during inference and resorts to those obtained from individual minibatches for the sake of stochasticity in predictions. Clearly, this indicates a fallback from the maximum attainable performance of a network, especially given the whole purpose of BReN. Also, it should be noted that MCBN requires the training data to be available for the sampling of minibatches for stochastic normalization during inference. Last but not least, MCBN further requires that the same minibatch size be specified during training and inference; otherwise, the respective approximate posteriors would be inconsistent [10]. Considering that inference generally takes less resources than training, one may try to use a larger batch size during inference; however, this is prohibited by MCBN, while permitted by BN. Also, if a *pre-trained* network is used for inference, it is virtually impossible to figure out the training minibatch size, unless it is documented somewhere.

The quality of uncertainties offered by Bayesian neural networks depends on the suitability of the prior and the approximation quality which is usually tied to computational constraints. *Non-Bayesian* alternatives can offer simpler yet effective means to quantify the uncertainties of deep neural networks. [6] has recently proposed an *ensemble* approach and combined it with *adversarial* training. Basically, the ensemble consists of multiple neural networks and it is diversified by their random initializations as well as random shuffling of training examples. As a result, it readily provides predictive distributions and uncertainty estimates during inference. Adversarial examples, which are essentially augmented training examples: $\mathbf{x}' = \mathbf{x} + \varepsilon \text{sign}\left(\nabla_x J(\theta, \mathbf{x}, y)\right)$ [15, 6], explore the localities of the original training examples to test the robustness of neural networks. As a result, the likelihood of the data smooths out in the $\varepsilon$-neighborhood of examples and the ensemble generates well-calibrated outputs [6]. Note that the exploration is guided by *gradients* and the adversarial examples are attracted towards the objective function (Figure 1). While the procedure cleverly avoids the computational burden of the ideal exploration in all $2^D$ directions in $\{+1, -1\}^D$ [6], it clearly leaves us with the following question: *Can we efficiently explore the localities of examples in all directions in order to obtain predictive uncertainty estimates?*

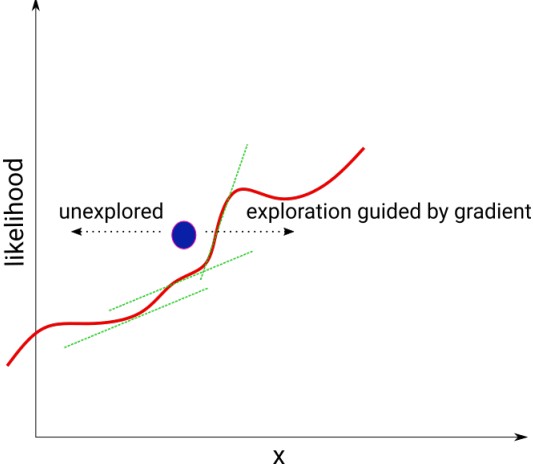

Figure 1: Illustration of $\varepsilon$-neighborhood exploration in 1-D. Given a data point (blue), adversarial examples generated by the *fast gradient sign method* are attracted towards the likelihood function (red) because of the positive gradients (green). Thus, the left hand side of the original example is left mostly unexplored.

# 3 Test-time Data Augmentation for Aleatoric Uncertainty Estimation

We aim to test whether it is possible to use data augmentation techniques as an efficient and effective means to explore the locality of the example given in Figure 1 with a sweeping capacity greater than that of adversarial examples generated by the fast gradient sign method [15, 6]. With more freedom to traverse different parts of input spaces, we aim to estimate the *heteroscedastic aleatoric* uncertainty of deep neural networks. To this end, we resort to *test-time* data augmentation and propose a general purpose uncertainty estimation method that is independent of training procedures as well as the choice of pretrained architectures. However, we emphasize the use of BReN for training in order to attain the maximum performance of a network.

Our approach amounts to generating $T$ augmented examples per test case and feeding them to a neural network so as to obtain a predictive distribution which can then be used to evaluate the predictive uncertainty. We conjecture that simple and randomized transformations of examples will enable us to efficiently and effectively estimate the input-dependent uncertainties of deep neural networks. Our data augmentation pipeline consists of the following geometric and color transformations in the given order: random crop and resize, random brightness, hue, saturation, and contrast adjustments, random horizontal and vertical flips, and random rotation.

# 4 Experiments

We evaluate our method using the well-known collection of fundus images obtained from a previous Kaggle competition [16]. The dataset consists of 35,126 training images and 53,576 test images graded by clinicians according to DR scales (Figure 2). During the competition, the labels of 10,906 test images were *public* and used for validation purposes by participants. Rest was *private* and internally used for ranking submitted solutions. The dataset is severely imbalanced. It is dominated by the examples of No DR, which corresponds to ∼73% of images in both the training and test sets.

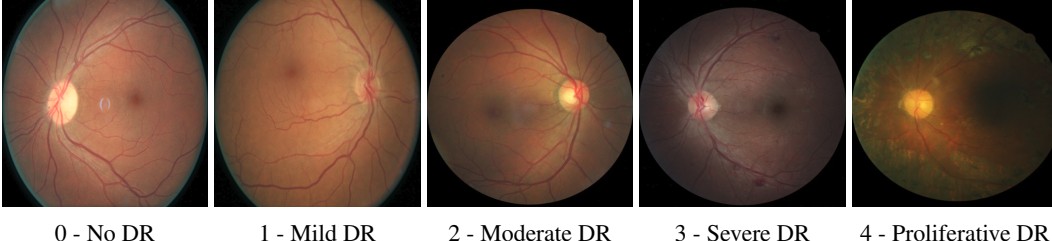

| 0 - No DR | 1 - Mild DR | 2 - Moderate DR | 3 - Severe DR | 4 - Proliferative DR |

Figure 2: Examples of fundus images from the Kaggle DR dataset. Class priors based on the training set are as follows: 0: 73.48%, 1: 6.95%, 2: 15.06%, 3: 2.48%, 4: 2.01%.

## 4.1 Image Processing and Data Augmentation

All images are cropped to a squared center region and resized to $512 \times 512$ pixels (Figure 2). Then, they are color-normalized as described in [7] for contrast enhancement. For simplicity and the sake of floating point representation of images, pixel values are mapped from $[0, 255]$ into $[0, 1]$. In addition to image processing, we perform data augmentation via the pipeline described in Section 3 during training. With a probability of 0.5, each image is cropped to a box, corners of which are randomly sampled from the margins of $1/3$ of height or width from each side, and resized to $512 \times 512$. Then, color transformations are applied with offsets for brightness and hue adjustments sampled from $[-0.5, 0.5)$ and $[-0.5, 0.5]$, respectively. Saturation and contrast factors are from $[0, 3]$. At this point, resulting pixel values are clipped into $[0, 1]$. Then, the remaining geometric transformations are applied. The probability of horizontal or vertical flip is 0.5 and the image is rotated by an angle from $[-\pi, \pi]$. Figure 3 shows several examples obtained via the pipeline. We use the same configuration at test time, as well.

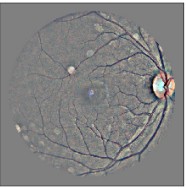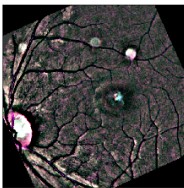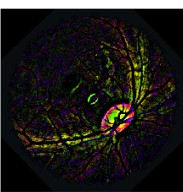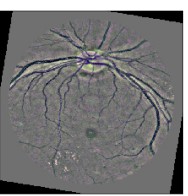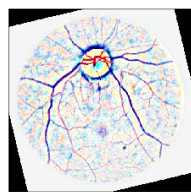

Figure 3: Exemplary images obtained via image processing and data augmentation. Images exhibit high diversity due to the wide range of parameter specifications used in the pipeline.

## 4.2 Network Architecture, Training and Inference

We implemented[1] a CNN based on the ResNet50 [13, 14] architecture, details of which are provided in Table 1. While we followed the principles of the original architecture, we made our own design choices, as well. For instance, our network includes an additional fully connected layer before softmax. Also, it uses parametric ReLUs (PReLUs) [17] in the first convolutional stack and the fully connected layer. Since these parts of the network has no residual connections, we used PReLUs in order to promote gradient propagation through those layers. The intermediate convolutional stacks fully enjoy the residual architecture with ReLUs and head nodes use max pooling to downsample their inputs in order to match the dimensions to the corresponding residual parts. The use of max pooling in lieu of 1x1 convolutions with strides of 2 aims to keep a *clean information path* with no scaling effects as per [14]. All weight layers in the network use BReN. Lastly, we concatenate the max and average pooled features from the convolutional stack before the fully connected layer.

We train the network with softmax cross-entropy loss in an *end-to-end* fashion for 1,000,000 iterations with Adam [18]. For the first 200,000 iterations, we use balanced minibatches of 5 in an attempt to combat severe class imbalance. Then, we increase the minibatch size to 8 and sample stratified batches until the end of training, which lets the network adjust itself to the true distribution of classes with an incentive of oversampling[2]. As a result of data augmentation, the network is exposed to 7.4 million images generated during training. Weights are initialized via *He's initialization* [17] and we use weight decay with $\lambda = 1e - 5$. Initial learning rate is 0.005 and it decays with a rate of 0.96 at every 25,000 iterations. Also, the performance of the network is validated on the public examples once in 10,000 iterations. The best performing configuration is saved and used for inference. When making predictions via test-time data augmentation, we generate $T = 128$ augmented examples and rely on the median probability scores. We evaluate the uncertainties w.r.t. interquartile range.

## 4.3 Results

In addition to the 5-way classification of fundus images, we introduce two additional binary classification tasks by considering two scenarios for *healthy vs. diseased* separation. Given an onset level, say 1 - Mild DR or 2 - Moderate DR, we dichotomize the network outputs by summing up the softmax values according to the onset level. Figure 4.a shows the training curve of the network. Zooming into the tail reveals that the network needs more training until convergence. Despite its non-convergence even after 1,000,000 iterations, its validation performances on two binary classification tasks (Figure 4.b) suggest that the network has reached a reasonable discriminative capacity. Under the onset level 1 scenario, it achieves a ROC-AUC score of 0.876, whereas Leibig et al. reported 0.889. Given that our priority is uncertainty estimation, we save the discriminative performance improvements for later. Further analysis of the performances via multi-class ROC-AUC scores in Figure 5.a reveals that the most of the network's discriminative power stems from its ability to recognize the examples of two classes: Class 0 (no DR) and Class 3 (severe DR). While the examples of Class 1 (mild DR) and 2 (moderate DR) have been shown to be the most difficult ones [7] as these are transitional stages of DR, the same network performs poorly on the examples of Class 4 (proliferative DR), which is a bit surprising since the patterns of proliferative DR are expected to be more pronounced in comparison

---

[1]Code available at https://github.com/berenslab/ttaug-midl2018
[2]Minibatches of 8 consists of 3,1,2,1 and 1 examples from each class, which still induces an oversampling of the minority classes; however, it preserves the majority of the No DR examples.

Table 1: Network architecture based on ResNet50 [13, 14] and implemented in Tensorflow 1.4.1 [19]

| Stack ID | Output size | Act. Func. | Specification | Note |
|---|---|---|---|---|
| input | $512 \times 512 \times 3$ | - | - | |
| conv1 | $256 \times 256 \times 64$ | PReLU | $5 \times 5, 64, /2$ | |
| | $128 \times 128 \times 64$ | - | $3 \times 3$ max pool, $/2$ | |
| conv2 | $128 \times 128 \times 256$ | ReLU | $\begin{bmatrix} 1 \times 1, 64 \\ 3 \times 3, 64 \\ 1 \times 1, 256 \end{bmatrix} \times 3$ | head block does not downsample |
| conv3 | $64 \times 64 \times 512$ | ReLU | $\begin{bmatrix} 1 \times 1, 128 \\ 3 \times 3, 128 \\ 1 \times 1, 512 \end{bmatrix} \times 4$ | head block downsamples by /2 |
| conv4 | $32 \times 32 \times 1024$ | ReLU | $\begin{bmatrix} 1 \times 1, 256 \\ 3 \times 3, 256 \\ 1 \times 1, 1024 \end{bmatrix} \times 6$ | head block downsamples by /2 |
| conv5 | $16 \times 16 \times 2048$ | ReLU | $\begin{bmatrix} 1 \times 1, 512 \\ 3 \times 3, 512 \\ 1 \times 1, 2048 \end{bmatrix} \times 3$ | head block downsamples by /2 |
| | $1 \times 1 \times 4096$ | | $16 \times 16$ max pool $+ 16 \times 16$ avg. pool | |
| fc | | PReLU | Fan in: 4096 Fan out: 1024 | |
| softmax | | *softmax* | Fan in: 1024 Fan out: 5 | |

to severe DR. Reminding ourselves that proliferative DR is the most underrepresented class in our dataset, we speculate that the network would have achieved better performance on such examples if it had converged to a better solution. Selection of better hyperparameters and improved training procedures can remedy this situation along with the slight overfitting that can be observed from Figure 5, which we also leave for further investigation.

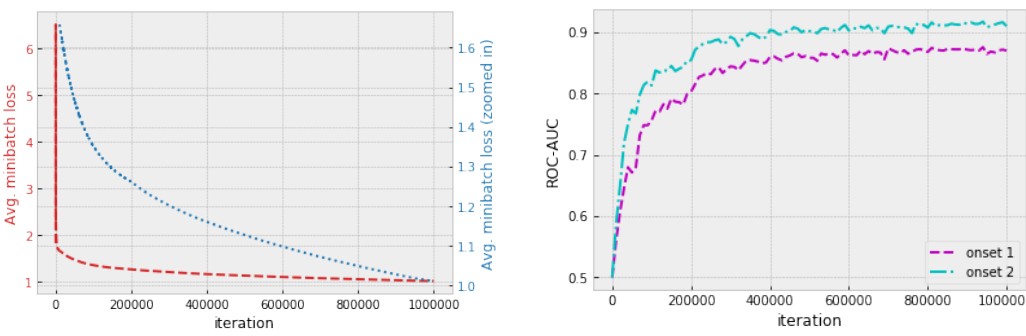

(a) Training curves through the entire training (red) and between the 10001-1,000,000th iterations (blue).

(b) Validation performance of the network w.r.t. different onset levels

Figure 4: Training and validation curves of networks.

Test-time data augmentation yields a competitive discrimination performance (Figure 5.b) and offers slight improvements on all of the baseline performance scores achieved via the traditional inference scheme. The performance improvement is a by-product of test-time data augmentation and our results suggest *error-correction* via neighborhood exploration. We find it promising for boosting the network's performance, explaining the decisions as well as relabeling examples, all of which will require further analysis. For now, in order to demonstrate the effectiveness of test-time data augmentation for uncertainty estimation, we consider two binary classification tasks introduced earlier.

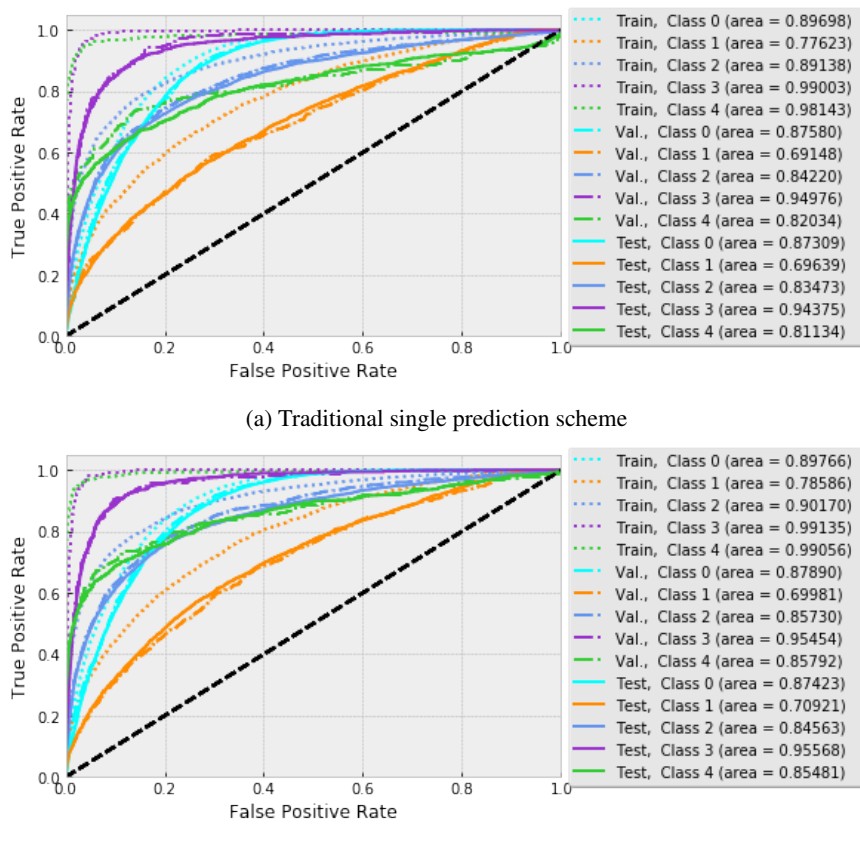

(a) Traditional single prediction scheme

(b) $T$ predictions via test-time data augmentation

Figure 5: Multi-class Receiver Operating Characteristics of the network.

Figure 6 shows the distributions of the input-dependent predictive uncertainties in both scenarios. Clearly, missed predictions are associated with higher uncertainty levels, which are also indicative of difficult cases that are usually located around class boundaries as shown by [7]. Similar to their case, we propose to leverage the uncertainty information in order to refer the difficult cases for further inspection by a physician. To this end, we rank the predictions by their uncertainty estimates and refer them based on various levels of tolerated uncertainty for further inspection. Then, we measure the ROC-AUC scores due to the predictions for the remaining cases. As a result of decision referral, we obtain *monotonic* performance improvements (Figure 7) on the remaining data that are safer to diagnose automatically. If 50% of data is referred for further inspection, our network achieves a score of 0.92 among the remaining data, whereas Leibig et al. reported 0.94 at the same referral rate. Also note that random referral has no such gain, which indicates that our uncertainty estimates are useful.

## 5  Discussion

We propose a simple and general purpose predictive uncertainty estimation method for deep neural networks. In particular, we demonstrate its effectiveness on a well-known collection of fundus images curated for DR detection. Despite its simplicity, test-time data augmentation is efficacious for evaluating the heteroscedastic aleatoric uncertainty of deep neural networks. Via a decision referral scenario that mimics a clinical work flow, we show that the uncertainty measures obtained by our method are useful. To the best of our knowledge, we are the first to demonstrate the use of test-time data augmentation for the evaluation of deep neural networks' predictive uncertainty. Considering that MCDO is phasing out due to the replacement of dropout with BN as well as MCBN's practical issues outlined in Section 2, test-time data augmentation is a promising avenue to explore, especially as an alternative to expensive Monte Carlo methods and approximate Bayesian inference. As an extension to this work, we will investigate the impact of components in our data augmentation pipeline on

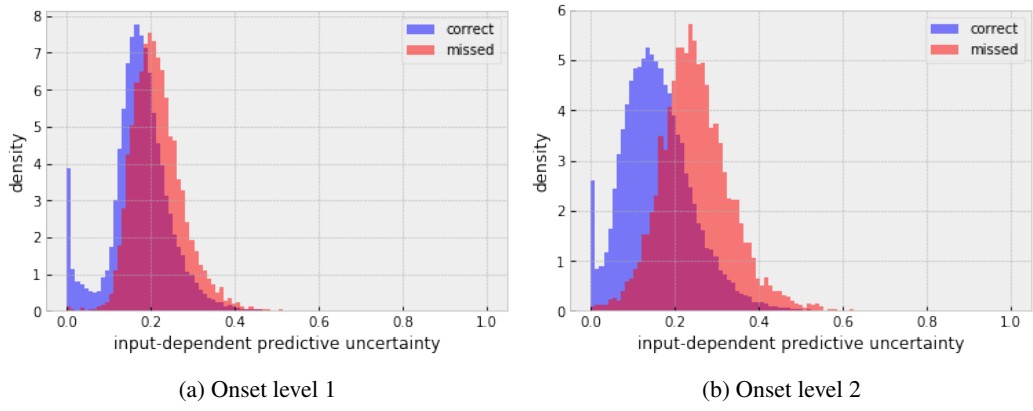

| (a) Onset level 1 | (b) Onset level 2 |

Figure 6: Distributions of uncertainties for all test images, grouped by classification results.

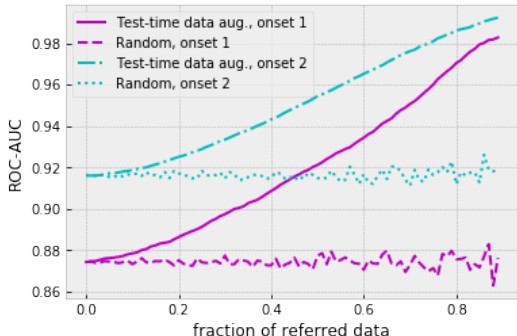

Figure 7: Improvement in performance via uncertainty-informed decision referral.

uncertainty and provide an analysis of the variation in input data. Also, we look forward to seeing what more sophisticated data augmentation methods will do to this end.

While the uncertainty estimates obtained via our method are useful, they may still reflect our lack of knowledge about the model parameters due to finite training data. In this regard, it will be essential to distinguish between the epistemic and aleatoric uncertainties. [20] has recently shown that *information gain* can do the trick. Adopting their approach, we will quantify how well our method captures the heteroscedastic aleatoric uncertainty of deep neural networks.

Ensemble approach to the estimation of predictive uncertainties is also of interest to us. Each member of an ensemble aims to reduce the bias and the ensemble uses their combined wisdom to tackle the variance. In this respect, we will construct an ensemble of networks. Under this scenario, we will follow the guidelines proposed by [6], exploit the randomness from parameter initialization and data shuffling as well as augmentation, and use our test-time data augmentation strategy to estimate predictive uncertainty. Given that the ensemble approach addressed the mode collapse of MCDO by better capturing the full posterior [20], we expect to improve the explanation of aleatoric uncertainty via ensembles of networks.

In the scope of our study, we focus on aleatoric uncertainty; however, epistemic uncertainty is also useful in small data scenarios and for detection of *out-of-sample* examples, e.g., *anomalies*. Even though the epistemic uncertainty is due to the model parameters, it will be interesting to see what the test-time data augmentation approaches will bring in this regard. We leave it for future work.

## Acknowledgments

We thank Christian Leibig for many useful discussions, helpful suggestions as well as providing the preprocessed images. We are also grateful for anonymous reviewers' efforts and suggestions. This research was supported by the German Ministry of Science and Education (BMBF, FKZ 01GQ1601) and German Science Foundation (BE5601/4-1).

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
