# OpenReview forum: "Test-time Data Augmentation for Estimation of Heteroscedastic Aleatoric Uncertainty in Deep Neural Networks"
_MIDL.amsterdam/2018/Conference — MIDL 2018 Oral_

### Review · AnonReviewer2 · 2018-05-08
**Interesting study, but some conclusions are too strong**

**Rating:** 3
**Confidence:** 2

**Review:**

The authors have addressed the topic of uncertainty in deep neural network, which I acknowledge to be an important topic of research. Analysis of uncertainty in deep learning models can improve the insight into the decision structure within the model and improve its performance. The authors discussed different aspects of uncertainty and focused in this paper on heteroscedastic aleatoric uncertainty, which is explained as variable noise in the input data. The provided solution for heteroscedastic aleatoric uncertainty is data augmentation at test-time, and the authors have chosen interquartile range (for the predictions of 128 augmentations for each image) as an indicator of uncertainty. From the results, we see that a difference in uncertainty was found for correct and incorrect classified cases. When cases with high uncertainty were removed the performance of the model improved.

It is hard to validate whether the authors have really captured the heteroscedastic aleatoric uncertainty, and we think that the statement on page 7 ‘test-time data augmentation enables us to evaluate the heteroscedastic aleatoric uncertainty of deep neural networks’ is too strong. No analysis of the variation in input data is provided nor are details given about the extent of the augmentation used a test-time (e.g. the absolute or relative changes in brightness). This information is useful for researchers that would like to continue with your work. Also, the effect of (the changes in) augmentation setting on the uncertainty results is missing. For example, would additional augmentation increase the differentiation in uncertainty between correct and incorrect predicted cases?

**Special Issue:**

No

---

### Review · AnonReviewer3 · 2018-05-09
**A well written paper describing a simple but novel method for uncertainty estimation**

**Rating:** 4
**Confidence:** 2

**Review:**

Overall I found this paper to be well-written and informative.  It addresses the problem of uncertainty estimation in neural network classification, which is an important and relevant subject.  The state of the art is clearly well understood and is outlined in adequate detail.  The technique for uncertainty measurement described in this paper, while simple, is also novel and demonstrates useful results with noted advantages over other state of the art methods. It is applied to a publicly available dataset of fundus images.
The paper is of sufficient quality for acceptance, however I note some limitations and possibilities for improvement:
 - The last two sentences of section 3 are the main description of the method and as such could be extended and detailed a bit more thoroughly at this point. The reader can extract more information from the end of section 4.2, but it should come earlier and in a more focused form.
 - Figure 5 needs further explanation in my view.  It is not clear to me why the authors put focus on the improved performance at test-time when testing on augmented samples. This is not the purpose of the method, nor is it made clear why it should be the case that performance is improved.
 - A minor issue is the non-optimal performance of the network (in general) compared to state of the art and the surprisingly poor performance on the most severe class of abnormality.  This does not detract from the main result but clarification would improve the work and give added confidence to the reader.
 - Figures 6 and 7 provide useful insight into the uncertainty measurements over the dataset however it would be nice if space permitted to see some representative examples of cases and the certainty values assigned.

**Special Issue:**

No

---

### Review · AnonReviewer1 · 2018-05-17

**Rating:** 4
**Confidence:** 3

**Review:**

Paper is using data augmentation to estimate the uncertainty of a CNN applied to retinal image segmentation. Nice idea, but nothing revolutionary. It is well written and technical sound with proper references

**Special Issue:**

No

---

### Comment · ~Murat_Seckin_Ayhan1 · 2018-06-10
**Authors' response to reviews and the final decision**

We thank the anonymous reviewers and organizers for their efforts as well as suggestions. We addressed the items put forward by the reviews; however, we had to be brief due to space issues and the organizers’ request that we should avoid substantial changes. We expect to share more insights into the method and discuss the results in greater detail at the conference in Amsterdam.

Hope to see you there!

---

### Decision · Program_Chairs · 2018-05-15
**Paper63 Acceptance Decision**

Oral